# Multimodal CEA-targeted fluorescence and radioguided cytoreductive surgery for peritoneal metastases of colorectal origin

Jan Marie de Gooyer [1,2,4 ✉], Fortuné M. K. Elekonawo [1,2,4], Andreas J. A. Bremers[2], Otto C. Boerman[1], Erik H. J. G. Aarntzen[1], Philip R. de Reuver[2], Iris. D. Nagtegaal [3], Mark Rijpkema[1,5] & Johannes H. W. de Wilt [2,5]

In patients with colorectal peritoneal metastases scheduled for cytoreductive surgery, accurate preoperative estimation of tumor burden and subsequent intraoperative detection of all tumor deposits remains challenging. In this study (ClinicalTrials.gov NCT03699332) we describe the results of a phase I clinical trial evaluating [$^{111}$In]In-DOTA-labetuzumab-IRDye800CW, a dual-labeled anti-carcinoembryonic antigen (anti-CEA) antibody conjugate that enables both preoperative imaging and intraoperative radioguidance and fluorescence imaging. Primary study outcomes are safety and feasibility of this multimodal imaging approach. Secondary outcomes are determination of the optimal dose, correlation between tracer uptake and histopathology and effects on clinical strategy. Administration of [$^{111}$In]In-DOTA-labetuzumab-IRDye800CW is well-tolerated and enables sensitive pre- and intraoperative imaging in patients who receive 10 or 50 mg of the tracer. Preoperative imaging revealed previously undetected lymph node metastases in one patient, and intraoperative fluorescence imaging revealed four previously undetected metastases in two patients. Alteration of clinical strategy based on multimodal imaging occurred in three patients. Thus, multimodal image-guided surgery after administration of this dual-labeled tracer is a promising approach that may aid in decision making before and during cytoreductive surgical procedures.

[1] Department of Medical Imaging, Nuclear medicine, Radboud Institute for Health Sciences, Radboud University Medical Center, Nijmegen, the Netherlands. [2] Department of Surgery Radboud Institute for Health Sciences, Radboud University Medical Center, Nijmegen, the Netherlands. [3] Department of pathology, Radboud Institute for Health Sciences, Radboud University Medical Center, Nijmegen, the Netherlands. [4]These authors contributed equally: Jan Marie. de Gooyer, Fortuné M. K. Elekonawo. [5]These authors jointly supervised this work: Mark Rijpkema, Johannes H. W. de Wilt. ✉email: jan-marie.gooyer@radboudumc.nl

Approximately 3-20% of all patients with colorectal cancer will be diagnosed with peritoneal metastases (PM) at the time of diagnosis or during follow-up[1]. When metastatic disease is limited to the peritoneal surface, international standards recommend cytoreductive surgery combined with hyperthermic intraperitoneal chemotherapy (CRS-HIPEC) as a potentially curative treatment. This treatment improves survival of selected patients with colorectal PM compared to systemic chemotherapy regimens[2]. Successful treatment of patients with colorectal PM remains dependent upon several critical factors[2,3]. First, it is vital to carefully select those patients who might benefit from this extensive procedure because CRS-HIPEC is associated with significant morbidity[4]. CRS-HIPEC is only recommended when the peritoneal tumor load is not too extensive according to the peritoneal carcinomatosis index (PCI) or other scoring systems[5,6]. To determine the PCI, surgeons rely on intraoperative inspection and palpation, because conventional preoperative imaging methods such as computed tomography (CT) and [$^{18}$F]FDG-PET/CT usually fail to accurately estimate the extent of the disease[7,8]. As a result, futile laparotomy rates of 5–15% have been reported in patients who undergo surgery for peritoneal carcinomatosis[9–12]. Diagnostic laparoscopy (DLS) may be used as method to improve PCI assessment, but the additional predictive value is limited[13]. Therefore, more accurate imaging methods are needed for patients with PM who appear eligible for CRS-HIPEC and to identify patients with a low PCI. The second most important factor associated with survival is the completeness of cytoreduction[3,4]. Accurate discrimination between adhesions or fibrosis and tumor lesions can be difficult because surgeons can rely on visual and tactile information only. Improving the surgeons' ability to discriminate between malignant and benign tissue might increase the chances of achieving truly complete cytoreduction and improve long-term patient survival.

Both factors can potentially be overcome by using targeted multimodal imaging after administration of a dual-labeled tracer, which combines a gamma-emitting radionuclide and a NIR-fluorophore on the same targeting molecule. The radionuclide can be used for preoperative targeted SPECT/CT imaging and intraoperative radioguided localization of deeper seeded lesions[14]. Subsequently, the signal originating from the fluorophore can facilitate NIR-fluorescence-guided surgery; a powerful technique that facilitates real-time detection of small tumor deposits and differentiation between benign and malignant tissue[15–17]. After resection, the abdominal cavity and surgical resection sites can be assessed with NIR-fluorescence imaging to identify residual tumor tissue. This combination of radiodetection and NIR-fluorescence imaging offers a powerful synergy of preoperative imaging and intraoperative tumor detection.

A tumor marker that is considered highly suitable for targeting of colorectal cancer is carcinoembryonic antigen (CEA)[18,19]. CEA is overexpressed on the surface of colorectal cancer cells in more than 90% of all cases at a high antigenic density, while the expression on healthy tissue is 60 times lower on average[20]. Labetuzumab is a clinical grade humanized monoclonal antibody that specifically targets CEA. It has been extensively investigated as therapeutic agent, radiotracer and antibody-drug conjugate for the treatment of various malignancies[21,22]. The dual-labeled version of labetuzumab, [$^{111}$In]In-DOTA-labetuzumab-IRDye800CW, has been tested as a multimodal imaging agent for colorectal cancer in several preclinical in vitro, in vivo and ex vivo studies with promising results[23,24]. Therefore, we conducted this clinical trial evaluating the safety and initial feasibility of preoperative SPECT imaging, intraoperative radiodetection and NIR-fluorescence-guided surgery after intravenous administration of different doses of [$^{111}$In]In-DOTA-labetuzumab-800CW in patients with peritoneal carcinomatosis of colorectal origin.

## Results

### Primary outcomes

*Summary of safety and patient characteristics.* Sixteen patients provided written informed consent for the study. One patient withdrew because of a rescheduled surgical procedure. Fifteen patients participated in the study and received an intravenous injection of 2 ($n = 5$), 10 ($n = 5$) or 50 ($n = 5$) mg DOTA-labetuzumab-IRDye800CW labeled with 100 MBq of $^{111}$In. The median age of participants was 64 (range 36–73) (Table 1). Temperature, blood pressure and hematologic markers for liver and renal function remained stable in the hours after injection (day 0), at time of the SPECT/CT (day 4/5) and until the day of surgery (day 5/6). One study-related CTCAE grade 1 allergic reaction was reported after tracer infusion in the 50 mg dose cohort. One possibly related mild adverse event was observed in patient 3, who experienced a transient increase of liver enzymes at day 24 that did not require intervention. Three serious adverse events were reported but these were all attributed to cytoreductive surgery and unrelated to study procedures: Patient 4 experienced an intra-abdominal arterial hemorrhage at day 15 and an anastomotic leakage at day 20, both requiring surgical intervention. Patient 7 experienced a paralytic ileus and required prolonged TPN and placement of a nasogastric tube. Table 2 summarizes all reported adverse events. The relative radiation dose was estimated to be 5.0 mSv for patients, which is considered an acceptable dose according to the ICRP 62[25]. For the surgeon, the total radiation dose was estimated to be 12 μSv for a surgical procedure of 6 h at day 6 after administration of the tracer.

*Multimodal imaging feasibility.* CEA-targeted SPECT/CT imaging was feasible after administration of 10 or 50 mg of the tracer but not after administration of 2 mg. Adequate and sensitive visualization of peritoneal tumor deposits was feasible in the 10 and 50 mg dose cohorts, but not after administration of 2 mg. The intraoperative fluorescent signal was weak or absent in patients who received 2 mg of the dual-labeled antibody.

### Secondary outcomes

*CEA-targeted SPECT/CT imaging.* All patients underwent SPECT/CT imaging 4 or 5 days after tracer injection. Fourteen patients underwent CRS-HIPEC 5 or 6 days after injection. One patient was treated with CRS-HIPEC 7 days after tracer injection due to a rescheduled surgical procedure. SPECT/CT imaging revealed metastases in 8 out of 10 patients of the highest two dose cohorts. The mean SPECT/CT PCI was 3 (SD 3), compared to a mean clinical (surgical) PCI of 10 (SD 6) ($p < 0.001$). Small peritoneal metastases (<10 mm) could not be detected or distinguished from benign tissue on CEA-targeted SPECT/CT, and it was not possible to provide an accurate estimation of the PCI based on preoperative SPECT/CT imaging. CEA-targeted SPECT/CT imaging did show larger lesions such as primary tumors, ovarian metastases, local recurrences, lymph node metastases and peritoneal metastases larger than 10 mm. Figure 1a, b shows preoperative SPECT/CT images of patient #7 with clear accumulation of [$^{111}$In]In-DOTA-labetuzumab-IRDye800CW in bilateral para-aortocaval nodes and in the left supraclavicular region.

*CRS-HIPEC.* Eleven patients received CRS-HIPEC treatment for peritoneal metastases, two were treated with resection of the primary tumor and CRS-HIPEC and two patients were treated with re-do CRS-HIPEC for recurrent peritoneal disease after CRS-HIPEC. Complete cytoreduction (CC0) was achieved in 11 cases. Incomplete cytoreduction and termination of the procedure (CC2) occurred in 4 cases. One patient was diagnosed with

**Table 1 clinical characteristics.**

| Patient | Sex | Preoperative treatment | Pattern of metastasis | Histology | CC# | Clinical PCI | Hospital Stay in days |
|---|---|---|---|---|---|---|---|
| *2 mg group* | | | | | | | |
| 1 | M | | Metachronous | Mucinous adenocarcinoma | CC0 | 11 | 22 |
| 2 | F | | Metachronous | Adenocarcinoma | CC2 | 18 | 13 |
| 3 | M | | Metachronous | Adenocarcinoma | CC0 | 16 | 10 |
| 4 | F | | Metachronous | Adenocarcinoma | CC0 | 14 | 23 |
| 5 | F | Neoadjuvant systemic therapy* | Synchronous | Adenocarcinoma | CC0 | 2 | 8 |
| *10 mg group* | | | | | | | |
| 6 | M | Neoadjuvant systemic therapy** | Synchronous | Adenocarcinoma | CC0 | 9 | 8 |
| 7 | F | | Synchronous | Adenocarcinoma | CC2 | - | 9 |
| 8 | F | Neoadjuvant systemic therapy* | Synchronous | Adenocarcinoma | CC0 | 7 | 9 |
| 9 | F | Neoadjuvant systemic therapy | Synchronous | Mucinous Adenocarcinoma | CC2 | 19 | 12 |
| 10 | F | | Metachronous | Adenocarcinoma | CC0 | 5 | 7 |
| *50 mg group* | | | | | | | |
| 11 | F | | Peritoneal relapse after previous CRS | Adenocarcinoma | CC0 | 3 | 10 |
| 12 | F | | Metachronous | Mucinous Adenocarcinoma | CC0 | 7 | 12 |
| 13 | M | | Metachronous | Adenocarcinoma | CC2 | 13 | 7 |
| 14 | F | | Peritoneal relapse after previous CRS | Mucinous adenocarcinoma | CC0 | 14 | 10 |
| 15 | F | | Synchronous | Adenocarcinoma | CC0 | 2 | 9 |

# CC = cytoreduction completeness,
* Patients were treated with neoadjuvant systemic treatment. ** Patient was treated with palliative systemic therapy but deemed resectable after a significant clinical response

retroperitoneal lymph node metastases, two were considered irresectable due to extensive central mesenteric disease, and one patient had extensive irresectable tumor invasion in the superior mesenteric vein. Despite preliminary termination of the surgical procedure in these 4 patients, intraoperative radiodetection and NIR-fluorescence imaging was successfully performed in all cases.

*Intraoperative tumor detection.* Only 17 out of 28 (61%) malignant lesions could be detected with NIR-fluorescence imaging in the 2 mg dose group. In the 10 mg group, 95% ($n = 16$) of all malignant lesions could be visualized and distinguished from benign tissue with NIR-fluorescence imaging intraoperatively. In the 50 mg group, all 25 malignant lesions (100%) could be detected and distinguished from benign tissue with NIR-fluorescence imaging intraoperatively. Examples of intraoperative NIR-fluorescence imaging are shown in Fig. 2. False positive fluorescent lesions were found in the 50 mg cohort (22%, $n = 7$) and the 10 mg cohort (16%, $n = 3$). Histopathological analysis revealed that one of the false-positive lesions in the 10 mg cohort contained granulocytic inflammatory processes with necrosis, fibrosis and inflammation. A second lesion contained fibrotic tissue and liver parenchyma, and the third lesion contained focal colitis. In the 50 mg dose cohort, four false-positives contained a granulocytic inflammatory process with necrosis, one contained CEA-positive acellular mucin, and the remaining two false-positive samples contained large numbers of macrophages due to the presence of foreign bodies after a sigmoid resection 6 weeks prior to cytoreductive surgery. The majority of these false-positive lesions did not express CEA, except for the lesions containing focal colitis and mucin. One tumor-positive lesion was not detected with NIR-fluorescence imaging in the 10 mg group. Immunohistochemical analysis revealed only single tumor cells in the deposit, as a result of preoperative systemic treatment. The accuracy of NIR-fluorescence imaging is shown in Table 3.

Intraoperative radionuclide detection was feasible and proved useful for detection of weak or non-fluorescent deeper seeded suspect lesions such as primary tumors, local recurrences, retroperitoneal lymph nodes, or tumor invasion in other organs such as the pancreas, duodenum and spleen. In one case (patient #13), gamma probe detection revealed a suspicious mass along the mesenteric vein that was confirmed to be tumor after pathological analysis. This lesion was only visible with fluorescence imaging after surgical exposure due to its deeper location.

**Back table and pathological analysis.** Back-table gamma counting of all resected peritoneal lesions in the surgical theater with the gamma probe revealed a TBR of 2.62 (SD 0.49) in the 10 mg, and 2.73 (SD 0.94) in the 50 mg group compared to a TBR of 1.91 (SD 0.43) in the 2 mg cohort (Fig. 3a). Differences were not statistically significant ($p = 0.2$). 70% ($n = 7$) of the ten false-positive fluorescent lesions in the 10 and 50 mg dose cohorts emitted a low radiosignal, with a mean TBR of 1.4 (SD 0.86). Only 30% of these lesions ($n = 3$) had a TBR above 2. Two of these lesions expressed CEA due to the presence of focal colitis or CEA-positive mucin. The third false positive fluorescent lesion consisted of liver tissue and did not express CEA. All specimens that were confirmed to contain tumor cells by the pathologist were included in fluorescence tumor to background measurements. A total of 47 histological slides were included in the final analysis and all slides exhibited clear overlap of the NIR-fluorescence signal and localization of CEA-expressing tumor cells. Calculated fluorescence tumor to background ratios were 1.78 (SD 0.31), 2.44 (SD 0.52) and 2.14 (SD 0.20) in the 2, 10 and 50 mg groups respectively, but these differences were not statistically significant ($p = 0.11$) (Fig. 3b). The average blood levels of [111In]In-DOTA-labetuzumab-IRDye800CW were 0.02 (SD 0.01), 0.008 (SD 0.005), 0.005 (SD 0.001) and 0.005 (SD 0.002) %ID/ml, at 3 h, 4 days, 5 days and 6 days after injection, respectively. The

**Table 2 Adverse events.**

| Patient | Protein dose | Days post injection | Adverse event | Intervention | Related | CTCAE | SAE |
|---|---|---|---|---|---|---|---|
| 1 | 2 mg | 11 | Fever (without focus) | Yes (antibiotics) | No | 2 | No |
| | | 28 | Delayed gastric emptying/ gastroparesis | Yes (nasogastric tube) | No | 3 | No |
| 2 | 2 mg | 16 | Seroma | Yes (aspiration) | No | 2 | No |
| 3 | 2 mg | 24 | Elevated liver enzymes (transient) | No | possible | 1 | No |
| 4 | 2 mg | 15 | Intraabdominal hemorrhage (arterial) | Yes (radiological coiling) | No | 3 | Yes |
| | | 20 | Anastomotic leakage | Yes (relaparotomy) | No | 3 | Yes |
| 6 | 10 mg | 10 | Chest pain | none | No | 2 | No |
| 7 | 10 mg | 7 | Paralytic ileus | Yes (total parenteral nutritionN & nasogastric tube) | No | 3 | Yes |
| 9 | 10 mg | 11 | Abdominal discomfort and fever | Yes (antibiotics) | No | 2 | No |
| 11 | 50 mg | 14 | Urinary retention | Yes (prolonged supra pubic catheter) | No | 2 | No |
| 13 | 50 mg | 0 | Mild allergic reaction (Rash) | no | yes | 1 | No |
| 14 | 50 mg | 7 | Hyperglycemia | Yes (insulin therapy) | no | 2 | No |
| | | 10 | Fever | No | no | 1 | No |
| | | 11 | Anemia (pre-existing) | Yes (2 packed cells) | no | 3 | No |

calculated biological half-life of the tracer was 59 (SD 16.4) hours (supplemental Fig. 1).

*Clinical strategy alterations.* Combined multimodal image-guided surgery resulted in alteration of clinical strategy in 30% ($n = 3$) of cases in the two feasible dose cohorts Previously undetected retroperitoneal lymph node metastases that were visible on SPECT/CT images in patient #7 were localized using radioguidance prior to cytoreduction. Frozen sections were taken from these corresponding lymph nodes under NIR-fluorescence guidance. SPECT/CT and corresponding images are visualized in Fig. 1. Immunohistochemistry confirmed the presence of CEA-expressing colorectal cancer cells. A total of four peritoneal lesions that were initially missed during CRS were detected with NIR-fluorescence imaging in two patients, resulting in a different PCI and alteration of surgical strategy after re-inspection of these suspect lesions. These lesions were small peritoneal lesions in the right paracolic region and pelvis that were confirmed to contain CEA-expressing colorectal cancer cells after histopathological analysis. Figure 4 shows intraoperative images and histopathological confirmation of an additional detected lesion in patient #15. Intraoperative detection of an additional lesion in patient 15 is shown in supplemental video 1.The protein dose of 10 mg was determined to be the optimal dose for imaging, due to the higher percentage of false positives found in the 50 mg group and the comparable detection rate of both dose levels.

## Discussion
This clinical trial showed that administration of [111In]In-DOTA-labetuzumab-IRDye800CW is safe and successfully facilitates radiodetection and NIR-fluorescence-guided surgery in patients with peritoneal metastases of colorectal cancer who are treated with CRS-HIPEC. The intravenous injection was well-tolerated, with only 2 mild related adverse events reported during the study period. Targeted intraoperative NIR-fluorescence and radioguidance after injection of [111In]In-DOTA-labetuzumab-IRDye800CW facilitated successful detection, localization and delineation of colorectal tumors, lymph node metastases and peritoneal metastases after injection of 10 or 50 mg of the tracer. These observations are supported by quantitative ex vivo tumor to background analyses of both the fluorescence and radiosignal. All but one of the 42 malignant lesions could be detected with

NIR-fluorescence imaging in the 10 and 50 mg cohorts and four additional (small) metastases that were initially missed during CRS were detected with intraoperative NIR-fluorescence imaging. The only false negative non-fluorescent lesion was found in the 10 mg cohort and consisted of a small group of single-cell clusters, surrounded by a thick fibrotic wall, hampering adequate NIR-fluorescence imaging. In this case ex vivo gamma counting confirmed tracer uptake (TBR 2.3), illustrating the complementary value of the radionuclide. A total of 30 clinically suspicious but true negative non-fluorescent lesions were resected at the two highest dose levels, providing a first indication that a non-fluorescent lesion can potentially be left in-situ to avoid unnecessary resections. Additional evaluation based on radiodetection with the gamma probe provides a secondary safeguard when determining if such lesions warrant surgical resection. After increasing the dose from 10 to 50 mg, the amount of false positive fluorescent lesions increased from 16% ($n = 3$) to 22% ($n = 7$), hampering the potential benefit of improved intraoperative tumor detection. Histopathological analyses showed that eight out of 10 false-positive fluorescent lesions did not express CEA and consisted of either foreign body reactions with high concentrations of macrophages or fibrotic inflamed tissue. Tracer uptake might be a result of the enhanced permeability and retention (EPR) effect that can be more pronounced in inflamed tissue. Another explanation might be that it is not always possible to accurately distinguish true positive fluorescence from autofluorescence caused by scar tissue or foreign body material, which has been reported in previous trials[16,17]. The latter is more likely, because the majority of false-positive fluorescent lesions exhibited a low radiosignal during ex vivo measurements.

This study shows that preoperative imaging of intraperitoneal metastases and subsequent calculation of the PCI based on CEA-targeted SPECT/CT imaging was not feasible. This could be the result of insufficient tracer accumulation in the small peritoneal tumors, and/or because of the limited spatial resolution of the imaging technique. Irresectable peritoneal disease was not detected with preoperative SPECT/CT imaging in three out of four patients in whom the procedure was aborted (CC2). However, preoperative SPECT/CT was feasible for detection of retroperitoneal and supraclavicular lymph node involvement that was not detected on standard preoperative imaging in one patient. This has clinical importance, because patients with PM and extraperitoneal or systemic disease are usually not eligible for

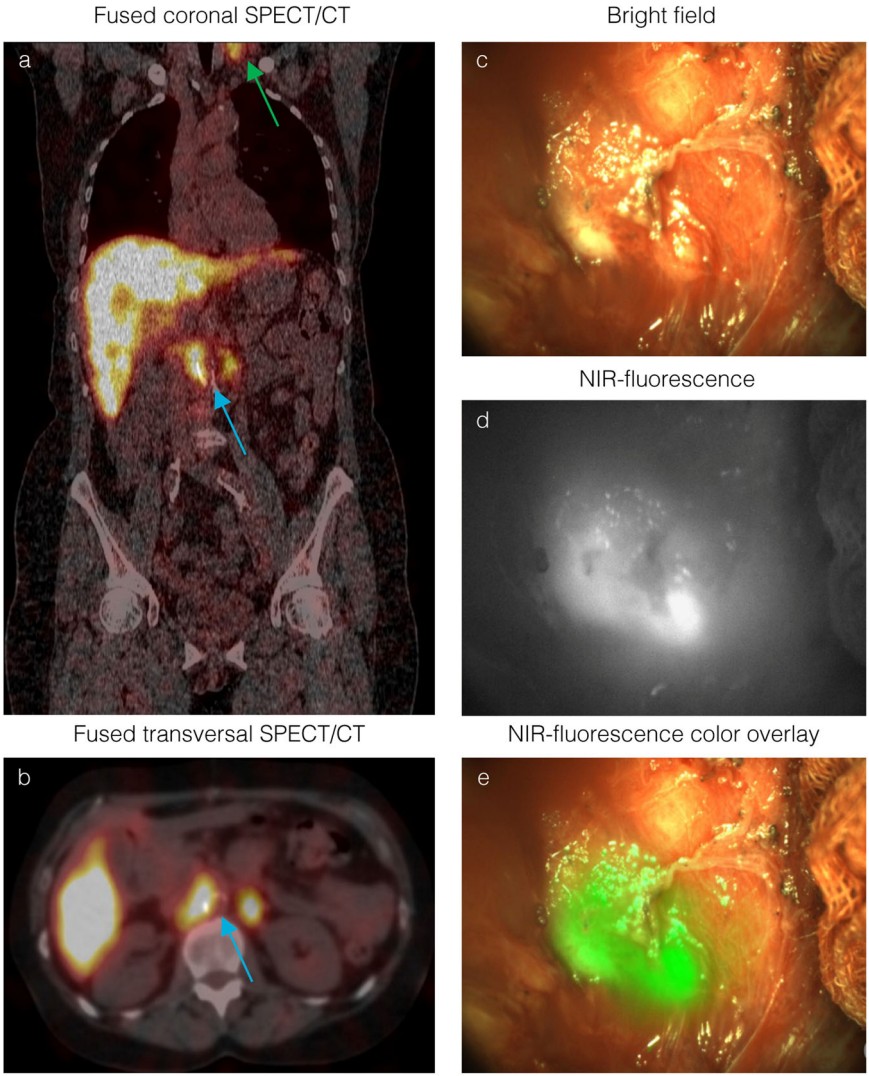

**Fig. 1 Multi-modal imaging of previously undetected colorectal lymph node metastases. a** (patient #7) Coronal section of preoperative CEA-targeted SPECT/CT images show accumulation of [$^{111}$In]In-DOTA-labetuzumab-IRDye800CW in retroperitoneal para-aortocaval (blue arrow) and left supraclavicular lymph nodes (green arrow). **b** Transversal section of retroperitoneal lymph nodes (blue arrow). **c** Intraoperative bright field, **d** NIR-fluorescence and **e** NIR-fluorescence overlay view of corresponding retroperitoneal lymph nodes show clear uptake of the tracer, corresponding with preoperative SPECT/CT imaging.

CRS-HIPEC. CEA-targeted SPECT/CT could thus be used as a potential tool to avoid a futile laparotomy and the associated morbidity. Similar patient selection based on CEA-targeted SPECT/CT imaging could be applied to other CEA-expressing gastrointestinal malignancies, for example to improve sensitivity and accuracy of response assessment after neoadjuvant chemoradiation in patients with rectal or esophageal cancer[26–29].

Since recent research has reaffirmed that complete surgical resection is still the cornerstone of CRS-HIPEC treatment, intraoperative imaging techniques such as radiodetection and NIR-fluorescence imaging could prove valuable tools for clinicians to achieve this[4]. Several NIR-fluorescence-guided surgery trials establishing the potential value of this technique have already been completed in patients with colorectal cancer. Harlaar et al. conducted a clinical pilot trial with a fluorescent VEGF-A targeting tracer in seven patients with colorectal PM scheduled for cytoreductive surgery[17]. They reported an excellent sensitivity but limited specificity (false positivity rate of 47%), most likely as a result of the more generally expressed target that they used. Boogerd and colleagues conducted a clinical trial with a

fluorescent anti-CEA tracer in patients with primary and metastatic colorectal cancer[16]. This trial provided clinical evidence of the additional benefit of CEA-targeted fluorescence-guided surgery, with a reported alteration of surgical strategy based on fluorescence imaging in 35% of all cases. However, the authors also address the limited tissue penetration of NIR-light, and the potential limitations when facing deeper seeded suspect lesions. he results of our study show that the multimodal imaging approach offers specific advantages over imaging with a single modality in selected cases, illustrated by the change in clinical management in 30% of cases based on the combination of radionuclide detection and NIR-fluorescence imaging. A potential limitation of CEA-targeted imaging is that high levels of serum CEA could result in complexing of the tracer in the circulation. However, Boogerd and colleagues already investigated the effect of circulating CEA on antibody availability for imaging[16]. They showed that only a small fraction of tracer is lost due to complexing with circulating CEA, even in patients with significantly elevated CEA levels (>100 mmol/liter). Taking into consideration that achieving complete cytoreduction remains the ultimate goal

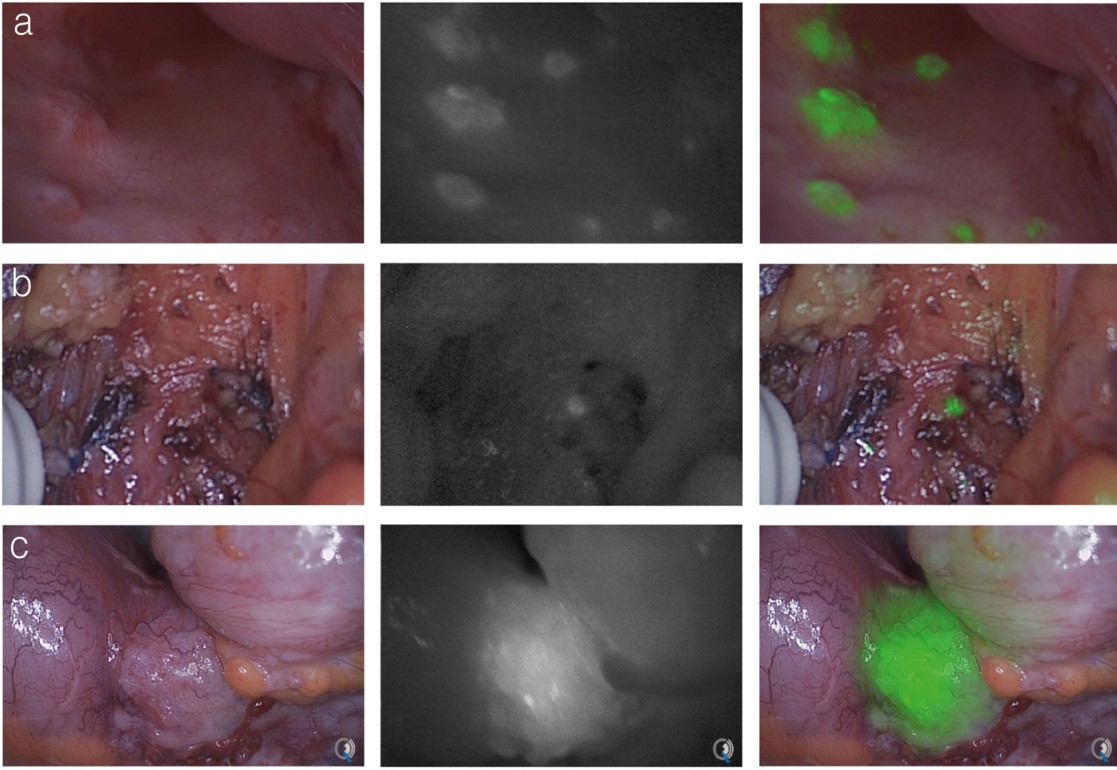

**Fig. 2 In-vivo fluorescence imaging of peritoneal metastases and a primary tumor. a** Small fluorescent hotspots in the pouch of Douglas of patient #9. **b** Intraoperative imaging after resection of suspected tissue reveals fluorescence signal in a small residual lesion of patient #6. **c** Fluorescence signal from a primary sigmoid tumor of patient #14. All fluorescent lesions were confirmed to contain CEA-expressing colorectal cancer cells on histopathological analysis.

| Table 3 Fluorescence detection rates. | | |
|---|---|---|
| | **Malignant** | **Benign** |
| *2 mg* | | |
| Fluorescence + | 17 | 0 |
| Fluorescence − | 11 | 10 |
| *10 mg* | | |
| Fluorescence + | 16 | 3 |
| Fluorescence − | 1 | 22 |
| *50 mg* | | |
| Fluorescence + | 25 | 7 |
| Fluorescence − | 0 | 8 |

during CRS-HIPEC, clinicians will continue to be confronted with substantial procedure-associated morbidity when performing extensive multi-visceral resections[30]. Only one false-negative non-fluorescent lesion was detected at the two feasible dose levels of our study, and this lesion was deemed malignant after gamma counting. The specificity and negative predictive value of multimodal image-guidance needs to be elucidated in a larger cohort, but these results provide an indication that surgeons could leave non-fluorescent lesions in situ, when using radiodetection as an additional safeguard. This could potentially result in less extensive cytoreductive procedures because 37% (*n* = 30) of 82 clinically suspect resected lesions did not contain malignant cells, and these lesions were not considered malignant based on NIR-fluorescence imaging and radiodetection. These findings are in agreement with literature describing the accuracy of clinical intraoperative assessment of the PCI[17,31].

This study has some strengths and pitfalls that need to be addressed. This is a phase I clinical study that investigates combined radiodetection and NIR-fluorescence imaging in patients with colorectal cancer. Due to the nature of this phase I study, it lacks power to reliably assess the diagnostic accuracy of these combined imaging techniques. Nevertheless, the study provides an indication regarding the potential feasibility and benefits, illustrated by the change of clinical management in three cases based on pre- and intraoperative imaging. Thus, our results justify the pursuance of a dose expansion study with the 10 mg dose to further investigate the diagnostic accuracy and clinical benefit of CEA-targeted multimodal image-guided surgery. The diagnostic imaging accuracy of preoperative imaging could potentially improve by choosing a positron-emitting radionuclide such as Zr-89 that enables PET imaging. However, subsequent accurate real-time intraoperative detection would not be feasible with this radioisotope, and surgeons would be exposed to much higher radiation dose levels compared to In-111. We recognize that some heterogeneity exists regarding the time interval between tracer injection and imaging timepoints, potentially hampering an accurate comparison due to differences in clearance and decay. Imaging proved feasible at all timepoints, but future studies are warranted to determine optimal timing between tracer injection and imaging. In addition, all resected lesions were measured by gamma counting ex-vivo, but intraoperative radionuclide detection was only performed in selected cases and for specific lesions. Radiodetection of lesions close to the liver was hampered by the background signal of the liver, however, NIR-fluorescence imaging of small lesions was feasible, even close to the liver. Finally, a pre- or co-dosing strategy with unconjugated antibody could potentially improve tumor penetration of the imaging conjugate even further[32]. Whether this applies

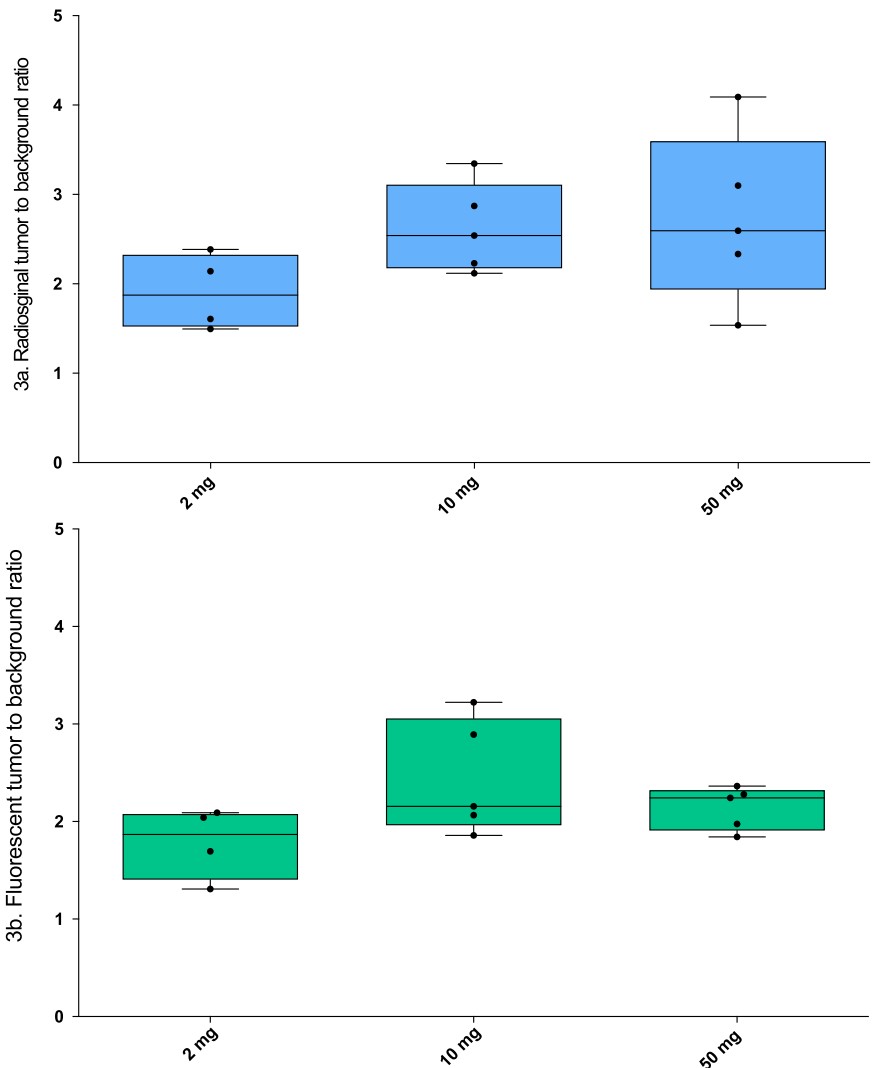

**Fig. 3 Ex-vivo radiosignal and NIR-fluorescence based tumor to background ratios. a** Ex-vivo radiosignal tumor to background ratios; radiosignal based tumor-to-background ratios based on back table gamma probe measurements is shown at all 3 dose levels. **b** Ex-vivo fluorescence tumor to background ratios, the NIR-fluorescence tumor to background ratios based on micropscopic NIR-fluorescence measurements of tissue sections is shown. The panels have also been described in the results section under the header: Back table and pathological analysis. Dose level 2 mg $n = 4$ subjects, 10 mg $n = 5$ subjects and 50 mg $n = 5$ subjects. Differences between TBR were not significant for both the radiosignal ($p = 0.2$) and the fluorescent signal ($p = 0.1$) (one-way ANOVA testing with post-hoc Bonferroni correction). Source data are provided as a Source Data file.

to CEA-targeting tracers and results in improved contrast in vivo needs to be investigated in future clinical trials.

In conclusion, this study demonstrates that CEA-targeted pre-operative SPECT/CT imaging and multimodal image-guided surgery after intravenous administration of [111In]In-DOTA-labetuzumab-IRDye800CW is safe and feasible in patients treated with CRS-HIPEC. Our multimodal imaging strategy resulted in preoperative detection of extra-abdominal disease in one patient and intraoperative detection of additional malignant lesions in two patients. A dose expansion study is currently being conducted to further elucidate the diagnostic accuracy and potential implications for patients with peritoneal metastases of colorectal cancer.

## Methods

### Preparation of [111In]In-DOTA-labetuzumab-IRDye800CW. Labetuzumab
10 mg·ml⁻¹ was provided by Immunomedics Inc. (Morris Plains, NJ, USA). [111In]In-DOTA-labetuzumab-IRDye800CW was produced under metal-free conditions and in compliance with Good Manufacturing Practice (GMP) in the radiopharmacy facilities of the Radboudmc. The manufacturing process was similar to the preparation of Girentuximab-DOTA-IRDye800CW 5 mg·ml⁻¹ which was described earlier.14 In short, labetuzumab 5 mg/ml⁻¹ was incubated with a 2–2.5-fold molar excess (to reach

IRDye800CW/IgG substitution ratio of 0.5–2.0) of IRDye800CW-NHS ester (LI-COR biosciences, Lincoln, NE, USA) followed by incubation with a 25-fold molar excess of the chelator DOTA-NHS (Macrocyclics, Dallas, TX, USA) (aiming at a DOTA/IgG substitution ratio 0.5–3.0) of the chelator DOTA-NHS (Macrocyclics, Dallas, TX, USA) in a 1.25 M NaHCO₃ buffer, pH 8.5. Hereafter, excess IRDye800CW-NHS ester and DOTA-NHS ester were removed by dialysis against 0.25 M ammoniumacetate, pH 5.5 containing Chelex 100 resin (Bio-Rad Laboratories Inc., Hercules, CA, USA) for 3 days to ensure metal-free conditions. After purification, the product was sterile filtered and aliquoted. DOTA-labetuzumab-IRDye800CW was stored at 4 °C in the dark until use. To confirm the CEA-specific binding of the dual-labeled antibody conjugate, the immunoreactive fraction of [111In]In-DOTA-labetuzumab-IRDye800CW was determined as described by Lindmo et al.31 Before each intravenous administration, the integrity of the dual-labeled antibody conjugate and radiochemical purity after radiolabeling was determined by high-performance liquid chromatography analysis. Before injection, 2, 10 or 50 mg DOTA-labetuzumab-IRDye800CW was labeled with 111In. In short, unlabeled DOTA-labetuzumab-IRDye800CW was incubated with 100-120 MBq [111In]InCl3 (Curium, Petten, The Netherlands) at 45 °C for 30 min in the dark. 100 MBq of the final product was diluted in 0.9% NaCl to a final volume of 10 ml and drawn up in a syringe shortly before injection.

**Subject population**. Patients were eligible for inclusion if they were 18 years of age or older, had histologically proven peritoneal metastases of colorectal cancer and were scheduled for CRS-HIPEC or redo CRS-HIPEC. Patients were excluded from participation if they were pregnant or breastfeeding, had a serum CEA concentration of

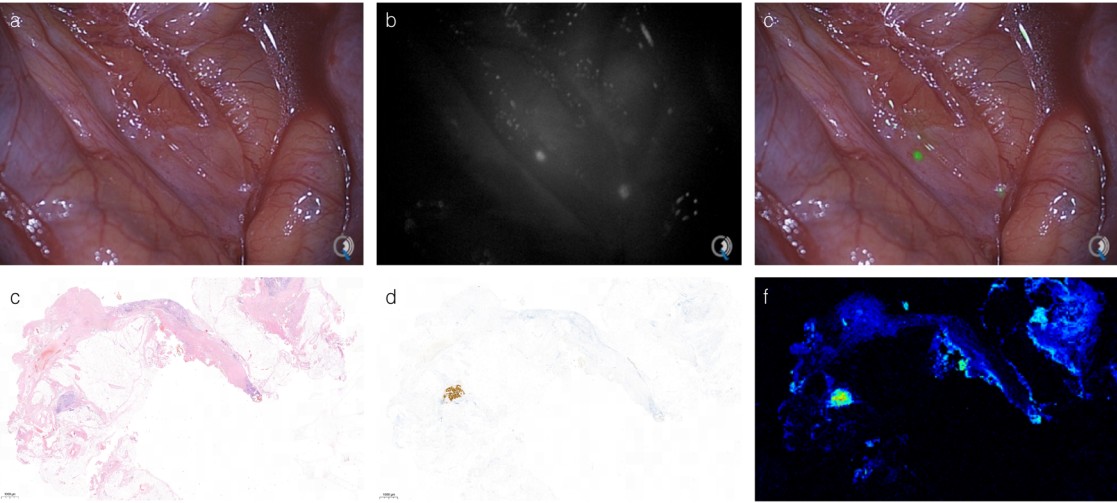

**Fig. 4 Intraoperative fluorescence detection of previously undetected peritoneal metastases.** (patient #15): Top row: Intraoperative NIR-fluorescence imaging of the right abdominal wall after cytoreductive surgery reveals two fluorescent lesions (**b**, **c**) not visible during standard visual inspection (**a**). Bottom row: pathological assessment of one of these lesions shows a CEA-expressing submillimeter tumor deposit on H&E (**c**) and CEA (**d**) immunohistochemistry that correlates with a strong fluorescence signal (**f**). Correlation between NIR-fluorescence and immunohistochemistry was performed on 2 or more different tissue sections for each independent case.

>500 ng/ml, had a known CEA negative malignancy, if a radionuclide had been administered within 10 physical half-lives prior to study enrollment or if coinciding uncontrolled medical conditions were present and deemed potentially compromising to the patient's wellbeing and the study. The study was registered at clinicaltrials.gov (ClinicalTrials.gov identifier: NCT03699332).The study was approved by the regional ethical review board (CMO region Arnhem-Nijmegen) and was performed according to the principles of the Declaration of Helsinki. Written informed consent was obtained from all patients prior to participation.

**Clinical trial design**

*Outcomes.* The primary objectives were to determine the safety and feasibility of multimodal image-guided surgery after intravenous administration of [111In]In-DOTA-labetuzumab-IRDye800CW in patients undergoing CRS-HIPEC for colorectal peritoneal carcinomatosis. Image-guided surgery was considered safe if no serious adverse events were attributed to [111In]In-DOTA-labetuzumab-IRDye800CW administration or study procedures. The study technique was considered feasible when specific accumulation of [111In]In-DOTA-labetuzumab-IRDye800CW in CEA-expressing tumor tissue could be detected with the available technologies (nuclear, optical and (immuno)histological). The optimal protein dose was considered the lowest dose that yielded feasible intraoperative imaging. Secondary outcomes were the number of patients in whom additional lesions were visualized by NIR-fluorescence imaging after cytoreductive surgery, the tumor to background ratio (TBR) as measured by ex vivo gamma measurements of surgical specimens and flatbed fluorescence measurements of 4 μm tissue sections, correlation of tracer accumulation and CEA expression, and to estimate the blood levels and clearance of [111In]In-DOTA-labetuzumab-IRDye800CW expressed as percentage injected dose per gram (% ID/g).

**Safety and pharmacokinetics**. This single-center, open label, phase I, dose escalation study was conducted to evaluate the safety, feasibility and optimal dose of [111In]In-DOTA-labetuzumab-IRDye800CW for pre- and intraoperative tumor detection in patients with colorectal peritoneal metastases. *Extensive toxicity tests have been performed for the mAb labetuzumab (hMN-14), as well as for the dye IRDye800CW[33] Together with the toxicity and safety tests performed for similar mAb conjugates (e.g. bevacizumab-IRDye800CW, cetuximab-IRDye800CW), and our own preclinical results with this specific tracer[23], the Dutch competent authorities waived the need for further toxicity testing[34,35]*Patients received a single intravenous dose of [111In]In-DOTA-labetuzumab-IRDye800CW five or six days prior to surgery and were monitored for adverse events until 3 hours after injection. Blood samples were drawn for safety and pharmacokinetic analysis at the following timepoints; 5 minutes before tracer administration, 180 min after injection, prior to SPECT/CT imaging (day 4 or 5), during cytoreductive surgery (day 5 or 6) and at the first outpatient clinical visit after discharge. Vital signs including heart rate, blood pressure, temperature and respiratory rate were measured before administration and 5, 30 and 180 min after injection. The final study follow-up visit coincided with the first post-surgery outpatient clinic visit. Treatment-related adverse events were defined as any adverse event associated with the study procedure but not necessarily related to the study intervention (i.e. [111In]In-DOTA-labetuzumab-IRDye800CW administration) for up to 10 days after surgery, using the National Cancer Institute Common Terminology Criteria for Adverse Events (version 4.03).

**Preoperative SPECT/CT imaging**. 4 or 5 days after administration of the dual-labeled antibody, SPECT/CT images of the thorax and abdomen were acquired with a dual-head Symbia T16 Truepoint SPECT/CT scanner (Siemens Healthcare, The Hague, The Netherlands). After acquisition of a low dose non-contrast enhanced CT, scintigraphic imaging was performed with the following settings: non-circular, angle 0-180°, 64 views per detector, 19 seconds per view with medium energy all-purpose parallel-hole collimators. Accumulation of [111In]In-DOTA-labetuzumab-IRDye800CW was scored by a nuclear medicine physician according to the PCI scoring method used intraoperatively as described by Sugarbaker[5].

**Image-guided surgery**. The surgical procedure was performed by a dedicated team of 2 experienced surgical oncologists and has been described in detail previously[36]. After surgical exposure of the abdomen, inspection and adhesiolysis, the PCI was scored and calculated as usual. Hereafter, the fluorescent PCI was scored using the dedicated NIR-fluorescence imaging system (QMI Spectrum NIR fluorescence camera, Quest Medical Imaging, Middenmeer, the Netherlands). Subsequently, standard of care cytoreductive surgery was performed. After completion of the procedure, the abdominal cavity including resection surfaces were systematically investigated using the gamma probe (Europrobe 3.2, SOE 3216-7, Eurorad SA, Eckbolsheim, France) and NIR-fluorescence camera. Residual fluorescent lesions were inspected and palpated, and only resected if the operating surgeon could establish that resection would not impose a negative effect on morbidity. Researchers performed back-table NIR-fluorescence imaging of all resected tissue specimens ex vivo. Back-table gamma probe detection of all suspected malignant lesions and one corresponding benign region was performed for quantification of tracer accumulation. Each resected lesion was noted on a form as clinically suspect for malignancy or not, maximum radiosignal in counts per second and fluorescent or not.

**Tissue analysis**. To enable 1-on-1 comparisons between intraoperative observations, ex vivo observations, and microscopic analyses, all samples were reviewed and annotated during the first steps of pathological processing (cassette allocations after formalin fixation). Surgical resection specimens underwent standard pathological processing and were evaluated and assessed by an experienced board-certified gastrointestinal pathologist to confirm tumor status on hematoxylin and eosin (H&E) 4 μm sections. For research purposes, 4 μm tissue slides cut from all FFPE blocks were scanned for the presence of fluorescent signal (800 nm) on the Odyssey CLx flatbed fluorescence imaging system (LI-COR Biosciences, Lincoln, NE, USA) and exported as tagged image file (TIFF) format using Image Studio Lite version 5.2.5 (LI-COR Biosciences, Lincoln, NE, USA). Sections were subsequently stained for H&E and CEA respectively and reviewed by an experienced board-certified gastrointestinal pathologist to verify the presence of colorectal tumor cells. Subsequently, regions of interest (ROIs) (tumor and adjacent normal tissue) were segmented on the H&E slides using caseviewer (CaseViewer 2.4, 3Dhistech, Budapest, Hungary) and subsequently imported in to Fiji-ImageJ (version 1.51) to measure mean fluorescence intensity (MFI) in tumor and benign tissue[37].

**Statistical analysis**. All clinical data was collected using Castor, Castor Electronic Data Capture 2019[38]. Because of the exploratory nature of this phase I study, no power calculation was done to determine the sample size. Descriptive statistics,

including mean radiosignal and fluorescence-based tumor to background ratios are depicted as mean ± SD. SPECT/CT-PCI and clinical(surgical)-PCI were depicted as mean ± SD and compared using a paired samples t-test. Statistical analyses were performed using Prism 5.03, GraphPad Software, (San Diego, CA, USA) and IBM SPSS Statistics 25.0, (Armonk, NY, USA). One-way ANOVA testing with post-hoc Bonferroni correction was performed to test for significant differences between the different protein dose levels in radiosignal-based and fluorescence-based tumor to background ratios. An alpha of 0,05 was used in all analyses and a p value <0.05 was considered statistically significant. The biological half-life of [$^{111}$In]In-DOTA-labetuzumab-IRDye800CW was calculated using mono-exponential regression analysis in MATLAB (Version R2018a, The MathWorks inc Natick, MA, USA)

**Reporting summary**. Further information on research design is available in the Nature Research Reporting Summary linked to this article.

## Data availability

Representative study data are presented in the manuscript and supplementary materials. Underlying data that support Fig. 3 and supplementary figure 1 are provided as a source data file. All Additional raw imaging and safety data that support the findings of this study are available from the corresponding author upon request. The redacted study protocol is available upon request from the authors. Source data are provided with this paper.

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

## Acknowledgements

We thank M. Hekman for her early drafts of the study protocol. We also thank D. Bos for her assistance in the lab and operating theater. The authors would like to thank our colleagues from the radiopharmacy department and hotlab for preparation of the dual-labeled antibody. We also would like to thank M. de Bakker for writing the SPECT/CT protocol. Lastly, we thank all involved technicians, surgeons, surgical nurses and the rest of the surgical team for their help during the procedures. This study was funded by the Dutch Cancer Society (grant number 10128). The content is solely the responsibility of the authors and the sponsor had no influence on study design and writing of the manuscript.

## Author contributions

J.M.D.G., F.M.K.E., A.J.B., M.R. and J.H.W. designed the trial, collected, analyzed and interpreted data, and wrote the manuscript. A.J.B., P.R.R. and J.H.W. performed the surgical procedures. O.C.B., I.D.N. and E.H.A. designed the trial and interpreted data. J.H.W. was the principal investigator. J.M.D.G., F.M.K.E. and M.R. have verified the underlying data. All authors reviewed the final version of the manuscript

## Competing interests

The authors declare no competing interests
