## [Peer Review File · Nature Communications]

REVIEWER COMMENTS

Reviewer #1 (Remarks to the Author):

This manuscript describes the phase I results of a well-known monoclonal antibody labetuzumab, already used in clinical trials using other radiolabeling methods, which targets CEA, but in the current trial labeled with two imaging moieties, IRDye800CW and [111In]In-DOTA. The novelty lies in the combination of fluorescent and radioguided surgery options of this approach, and the potential to perform radionuclide imaging pre-operatively.

The manuscript is well written.

The research approach is a phase I dose escalation study in 3 dose groups of each 5 patients, showing that the 2 highest doses are adequate for intra-operative use. The manuscript describes most of the relevant results in detail, but more detailed description of some parts would add to the paper.

Findings are very relevant to the field, and are promising for continued use of this approach in the cancer field.

In general, the materials and methods and results sections on SPECT/CT imaging should contain more details. Some points in the discussion would merit more in-depth discussion.

Please find my remarks below.

In figure 1, tracer accumulation clearly occurs in liver, but it is not discussed to what extent normal liver uptake of the tracer hampers correct assessment during surgery, for both radioguidance and fluorescent guidance. Please discuss.

The antibody labetuzumab has already been used in clinical trials in the past. Please also briefly discuss its current use as a therapeutic and discuss differences and (dis)advantages of labetuzumab as compared to other CEA antibodies used for imaging by other researchers (e.g. SGM-101).

Indeed, the limited spatial resolution of the imaging technique can explain why SPECT/CT did not meet the goal of correct pre-operative calculation of PCI. However, a similar tracer but with a more performant radioisotope might be able to overcome this problem, such as labeling with a PET-isotope such as F-18 or Zr-89. Please discuss this option.

No dosimetry has been performed for this tracer. Please discuss the expected dose to the patient and the medical staff during surgery.

What type of non-human toxicity testing has been done for the current conjugated antibody? No information is given in the manuscript and only limited details are provided in the protocol annex.

Discussion: "The results of our study show that the multimodal imaging approach offers specific advantages over imaging with a single modality in selected cases, illustrated by the change in clinical management in 30% of cases based on the combination of radionuclide detection and NIR-fluorescence imaging."

From the results, it is unclear in how many subjects /lesions, the multimodal imaging provided an added value over fluorescence-only imaging. This should be better detailed in the results section, or if not available, be discussed as a limitation of the current study.

Discussion: "Only one false-negative non-fluorescent lesion was detected at the two feasible dose levels of our study, and this lesion was deemed malignant after gamma counting." It is however not discussed if radioguidance before resection could have identified this lesion as malignant, as a higher background is typically present within the abdominal cavity (e.g. liver uptake). If such data is available, please add it to the results section. If not, please acknowledge this in the discussion section.

One of the limitations to this approach is circulating CEA, to which the antibody can bind and thereby remain in the blood. It is not clear if this was present in the patients included in the trial, and to what extent. Also correlating the biological half-life of the tracer to the level of circulating CEA could help understand this potentially limiting factor. It would add to the paper to discuss this further.

Materials and methods, SPECT/CT imaging:

"Whole body anterior and posterior SPECT/CT images": SPECT/CT images are obtained circularly around the body and not anterior / posterior, and moreover, are seldomly whole body (head to toe). Please rephrase to correctly reflect the acquisitions. Were planner images obtained?

"Accumulation ... was scored by a nuclear medicine physician": how was this scored? Were in the manuscript can these scores be found? In the results section, the scoring of SPECT/CT is only explained as 'feasible', which for me is not a score, but already an interpretation of such score. Please also provide real scoring data of SPECT/CT images, and how this was obtained. A table describing the number of lesions per size interval, and of the number detected in SPECT/CT imaging per subject could help clarify this. How were such results compared with PCI? Please provide more details.

In how many patients were metastatic lesions detected on SPECT/CT?

Representative images of subjects in the 3 different dose groups would add to help understand changes in general biodistribution due to mass effect.

Minor comments :

Abstract, p2 : occult undetected metastases : are occult metastases not always undetected?

Table 1: please explain abbreviations used in the table below the table. It is unclear what CC means.

Please add units to the column 'hospital stay', which I assume is expressed in days?

Table 2: column intervention mentions Yes /No or none for some, but immediately the intervention for other AEs. Please adapt for consistency. Also check use of capitals throughout the table.

please explain abbreviations used in the table below the table.

Figure 1, 2 and 3: please mention the patient number in the legend, to clarify patient characteristics and administered dose.

The text as well as the legend of figure 3 mentions figure 3a (or [A]) but in figure 3, no panel a is indicated. Moreover, the reference in the text to figure 3a does not appear for figure 3 but rather figure 4a?

Figure 3: please also explain what the second fluorescent lesions showed.

Reviewer #2 (Remarks to the Author):

Multimodal CEA-targeted fluorescence and radioguided cytoreductive surgery for peritoneal metastases of colorectal origin

Intraoperative tumor detection is a major question in our surgical field of oncologic peritoneal surgery, because able to fail and requiring time consuming and major expertise. Many of the surgeons believe that if we could be able to improve such detection, our results will be better and patient would survival longer.

For that reason, the subject of the paper is at the Journal's level.

Major Comments:

I have no major comment. The paper is very well writhed.

Minor comments:

i) Did the authors evaluated the duration of the procedure. The time necessary to perform the all abdominal cavity exploration ?

ii) Did the authors have been faced in some case were the finger identified something in a depth surface that is not identified by the light because of the depth ?

iii) Is the procedure feasible under laparoscopic surgery ?

iv) On results page 6 : "In the 10 mg group, 95% (n=16) of all malignant lesions could be visualized with NIR-fluorescence imaging" please complete information as visualization is per operative or on the back table ?

v) Sensitivity and specificity had to be reported on the abstract and on the paper, regarding per operative tumor detection.

vi) Sensitivity and specificity of the surgeon finger detection had to be reported to , as in the discussion part "This could potentially result in less extensive cytoreductive procedures because 37% (n=30) of 82 clinically suspect resected lesions did not contain malignant cells".

One idea ?

i) Include normal cases – as patient operated for a small colon cancer lesion to be certain that normal tissues did not have auto fluorescence. The antibody could have a spatial distribution affected by cancer location.

That question is important because authors reported false positive at 20% of cases, on the result section.

That paper is very interesting and had to be published

¹Point-to-point response to the reviewer comments

Reviewer #1 (Remarks to the Author):

This manuscript describes the phase I results of a well-known monoclonal antibody labetuzumab, already used in clinical trials using other radiolabeling methods, which targets CEA, but in the current trial labeled with two imaging moieties, IRDye800CW and [¹¹¹In]In-DOTA. The novelty lies in the combination of fluorescent and radioguided surgery options of this approach, and the potential to perform radionuclide imaging pre-operatively. The manuscript is well written.

The research approach is a phase I dose escalation study in 3 dose groups of each 5 patients, showing that the 2 highest doses are adequate for intra-operative use. The manuscript describes most of the relevant results in detail, but more detailed description of some parts would add to the paper.

Findings are very relevant to the field, and are promising for continued use of this approach in the cancer field.

We would like to express our gratitude towards the reviewer for his/her kind words.

In general, the materials and methods and results sections on SPECT/CT imaging should contain more details. Some points in the discussion would merit more in-depth discussion.

Please find my remarks below.

In figure 1, tracer accumulation clearly occurs in liver, but it is not discussed to what extent normal liver uptake of the tracer hampers correct assessment during surgery, for both radioguidance and fluorescent guidance. Please discuss.

Intraoperatively, we do observe a significant background signal originating from the liver due to hepatic clearance of the dual-labeled tracer. In practice, this may hamper radiodetection, but it does not hamper intraoperative fluorescence imaging of peritoneal metastases close to the liver, for example on the right hemidiaphragm or the peritoneum covering the liver itself. Radiodetection of lesions on the capsule of the liver however was not feasible due to high background signal. This is to be expected because the tissue penetration of gamma radiation is much bigger than that of the fluorescent signal. We have added this to the discussion section (lines 319-320).

The antibody labetuzumab has already been used in clinical trials in the past. Please also briefly discuss its current use as a therapeutic and discuss differences and (dis)advantages of labetuzumab as compared to other CEA antibodies used for imaging by other researchers (e.g. SGM-101).

One advantage of our tracer is that the fluorophore IRDye800CW emits light in the 800 nm range whereas SGM-101 emits light in the 700 nm range. At 800 nm, there is less autofluorescence, higher tissue penetration and less scattering compared to 700 nm. Furthermore, labetuzumab is a humanized antibody whereas the antibody used for the SGM-101 conjugate is chimeric. Humanized antibodies have the distinct advantage of having a longer circulating time and being less immunogenic because the immune response is directed against the murine section of the chimeric antibody.

We adapted the part of the introduction that describes labetuzumab to include its previous uses as a single drug agent, imaging tracer and antibody drug conjugate (lines 90-).

Indeed, the limited spatial resolution of the imaging technique can explain why SPECT/CT did not meet the goal of correct pre-operative calculation of PCI. However, a similar tracer but with a more performant radioisotope might be able to overcome this problem, such as labeling with a PET-isotope such as F-18 or Zr-89. Please discuss this option.

We do acknowledge that a positron-emitting radionuclide with a favorable half-life (Zr-89 would be ideal) would enable PET/CT imaging, which has a higher resolution. However, intraoperative detection of Zr-89 is not feasible because of the high energy 511 keV photons. Another drawback of using Zr-89 is that the absorbed radiation dose for surgeons would be significantly higher and could even exceed the maximum allowed annual exposure for professionals. We added these considerations to the limitations section of the discussion (lines 308-312).

No dosimetry has been performed for this tracer. Please discuss the expected dose to the patient and the medical staff during surgery.

Before the study started, we have estimated the expected dose to the patient and the surgeon. For the patient, the mean effective radiation dose after administration of a ^{111}In -labeled monoclonal antibody is 0.22 mSv/MBq, so the effective radiation dose of 100 MBq ^{111}In -labeled labetuzumab-IRDye800CW will be 22 mSv. Together with the low dose SPECT/CT, the total effective radiation dose will be 24.8 mSv. Thus, the relative radiation dose will be 5.0 mSv for patients aged >50y. This is considered to be an acceptable dose according to the ICRP 62 (3).

Surgery is performed 6 or 7 days after administration of ^{111}In -labeled labetuzumab-IRDye800CW. Considering the biological half-life of mAb's and the half life of ^{111}In , there will be 4.1 MBq of radioactivity left at day 6. Thus, the total radiation dose for the surgeon is 12 μSv per surgical procedure of 6 hours (assuming 0.087 $\mu\text{Sv}/\text{h}$ per MBq/m²). Even with a maximum of 10 procedures per year this is well within limits.

We have briefly added this to the manuscript (results section, lines 123-125).

What type of non-human toxicity testing has been done for the current conjugated antibody? No information is given in the manuscript and only limited details are provided in the protocol annex.

Extensive toxicity tests have been performed for the mAb labetuzumab (hMN-14), as well as for the dye IRDye800CW¹ Together with the toxicity and safety tests performed for similar mAb conjugates (e.g. bevacizumab-IRDye800CW, cetuximab-IRDye800CW), and our own preclinical results with this specific tracer², the Dutch competent authorities waived the need for further toxicity testing^{3,4}

Discussion: "The results of our study show that the multimodal imaging approach offers specific advantages over imaging with a single modality in selected cases, illustrated by the change in clinical management in 30% of cases based on the combination of radionuclide detection and NIR-fluorescence imaging."

From the results, it is unclear in how many subjects /lesions, the multimodal imaging provided an added value over fluorescence-only imaging. This should be better detailed in the results section, or if not available, be discussed as a limitation of the current study.

Combined detection revealed undetected tumors in two patients (#7 and #13), as described on line 139 and 178. However, in case 13 this did not result in alteration of surgical strategy because the patient was already considered to have irresectable disease. The 30% therefore refers to the 3 patients where imaging (either radiodetection or fluorescence) resulted in alteration of clinical strategy. We rephrased some sentences in the results section to provide more clarity.

Discussion: "Only one false-negative non-fluorescent lesion was detected at the two feasible dose levels of our study, and this lesion was deemed malignant after gamma counting." It is however not discussed if radioguidance before resection could have identified this lesion as malignant, as a higher background is typically present within the abdominal cavity (e.g. liver uptake). If such data is available, please add it to the results section. If not, please acknowledge this in the discussion section.

Unfortunately we did not perform intraoperative radiodetection of all malignant lesions and therefore can't report this specific data. Intraoperative radioguidance was only performed when fluorescence imaging was inconclusive or not feasible due to location of certain lesions. We have added this as a limitation to the discussion section.

We agree that higher background in the abdominal cavity can significantly hamper radioguided detection of tumor deposits, especially of smaller lesions. However, we also want to note that ex-vivo radiodetection could still potentially influence intraoperative decision making, as mentioned in the text.

One of the limitations to this approach is circulating CEA, to which the antibody can bind and thereby remain in the blood. It is not clear if this was present in the patients included in the trial, and to what extent. Also correlating the biological half-life of the tracer to the level of circulating CEA could help understand this potentially limiting factor. It would add to the paper to discuss this further.

Another clinical trial investigating CEA-target fluorescence-guided surgery with SGM-101 has already investigated the effect of circulating CEA on antibody availability for imaging (ref 16). They showed that only a small fraction of tracer is lost due to complexing with circulating CEA, even in patients with significantly elevated CEA levels (>100 mmol/liter). We added this to the discussion section (lines 283-).

Materials and methods, SPECT/CT imaging:

"Whole body anterior and posterior SPECT/CT images": SPECT/CT images are obtained circularly around the body and not anterior / posterior, and moreover, are seldomly whole body (head to toe). Please rephrase to correctly reflect the acquisitions. Were planar images obtained?

"Accumulation ... was scored by a nuclear medicine physician": how was this scored? Were in the manuscript can these scores be found? In the results section, the scoring of SPECT/CT is only explained as 'feasible', which for me is not a score, but already an interpretation of such score. Please also provide real scoring data of SPECT/CT images, and how this was obtained. A table describing the number of lesions per size interval, and of the number detected in SPECT/CT imaging per subject could help clarify this. How were such results compared with PCI? Please provide more details.

In how many patients were metastatic lesions detected on SPECT/CT?

Representative images of subjects in the 3 different dose groups would add to help understand changes in general biodistribution due to mass effect.

We rephrased the method section to correctly reflect the acquisitions that were used. We rephrased the methods section to provide more clarity on scoring.

We originally did not add the SPECT/CT-based PCI scores because of the low accuracy of this method, and phrased it as: it was not possible to provide an accurate estimation of the PCI based on preoperative SPECT/CT imaging. We now added several sentences to the results section where we compare the mean SPECT/CT PCI to the clinical PCI with a paired samples t-test. This clearly shows that the SPECT/CT PCI is significantly lower compared to the clinical PCI, reflecting that SPECT/CT imaging is not accurate in determining the intra-abdominal disease extent. We also

added a sentence to show in how many patients we detected metastases on SPECT/CT imaging (132-133).

Minor comments :

Abstract, p2 : occult undetected metastases : are occult metastases not always undetected?

We agree with the reviewer and have rephrased this.

Table 1: please explain abbreviations used in the table below the table. It is unclear what CC means.

Please add units to the column 'hospital stay', which I assume is expressed in days?

We have corrected both suggestions. CC is an abbreviation for Cytoreduction Completeness, the common scoring system used in peritoneal oncology: CC0: no residual disease; CC1: residual nodules measuring less than 2.5 mm; CC2: residual nodules measuring more than 2.5 mm.

Table 2: column intervention mentions Yes /No or none for some, but immediately the intervention for other AEs. Please adapt for consistency. Also check use of capitals throughout the table.

please explain abbreviations used in the table below the table.

We checked and adapted this. Abbreviations were removed and replaced by complete terminology.

Figure 1, 2 and 3: please mention the patient number in the legend, to clarify patient characteristics and administered dose.

We have adapted this.

The text as well as the legend of figure 3 mentions figure 3a (or [A]) but in figure 3, no panel a is indicated. Moreover, the reference in the text to figure 3a does not appear for figure 3 but rather figure 4a?

The reviewer is correct and we apologize for this error. The text has been adapted, and now refers to figure 4a and 4b. The legend of figure 3 has also been adapted.

Figure 3: please also explain what the second fluorescent lesions showed.

This lesion also contained colorectal cancer cells, but for the clarity and conciseness of the image we decided to only show immunohistochemical confirmation of one of these lesions. We adapted this in the legend.

Reviewer #2 (Remarks to the Author):

Multimodal CEA-targeted fluorescence and radioguided cytoreductive surgery for peritoneal metastases of colorectal origin

Intraoperative tumor detection is a major question in our surgical field of oncologic peritoneal surgery, because able to fail and requiring time consuming and major expertise. Many of the surgeons believe that if we could be able to improve such detection, our results will be better and patient would survival longer.

For that reason, the subject of the paper is at the Journal's level.

Major Comments:

I have no major comment. The paper is very well writhed.

We would like to express our gratitude towards the reviewer for his/her kind words.

Minor comments:

i) Did the authors evaluated the duration of the procedure. The time necessary to perform the all abdominal cavity exploration ?

Yes, imaging and systematic scoring of the fluorescent PCI (with additional radioguidance if necessary) took approximately 20 to 30 minutes. Back table evaluation did not influence procedure duration because this was done by researchers and not the surgeons.

ii) Did the authors have been faced in some case were the finger identified something in a depth surface that is not identified by the light because of the depth ?

Yes, but not very often. In general almost all superficial lesions could be detected with fluorescence imaging, even if they were covered by a thin layer of normal tissue, for example the retroperitoneal lymph node metastases visualized in Figure 1.

One example where fluorescence imaging was not feasible is a case where the surgeon palpated a suspicious lesion located near the head of the pancreas. No tumor lesion was visible on normal inspection and with fluorescence imaging. However, gamma probe detection revealed a high radiosignal of > 600 CPS. After surgical exploration, tumor ingrowth around the mesenteric trunc was discovered, which was also visible on fluorescence imaging, but only after surgical exposure of the mesenteric radix.

iii) Is the procedure feasible under laparoscopic surgery ?

Yes, several laparoscopic systems are capable of NIR-fluorescence imaging of fluorophores with a 800 nm wavelength. For example the Da Vinci robotic system with the Firefly fluorescence module, and the Olympus laparoscopy system. There are also gamma probes available for laparoscopy, and drop-in gamma probes that can be operated with the robotic forceps of the Da Vinci surgical robot.

iv) On results page 6 : "In the 10 mg group, 95% (n=16) of all malignant lesions could be visualized with NIR-fluorescence imaging" please complete information as visualization is per operative or on the back table ?

Intraoperative detection is reported. If a lesions was not fluorescent in vivo but fluorescent ex vivo than it was scored as negative. We clarified this in the text (lines 160-). Only if a lesion was resected prior to fluorescent scoring of the PCI, which sometimes happens during opening of the abdominal cavity or adhesiolysis, then the score is based on ex vivo imaging.

v) Sensitivity and specificity had to be reported on the abstract and on the paper, regarding per operative tumor detection.

vi) Sensitivity and specificity of the surgeon finger detection had to be reported to , as in the discussion part "This could potentially result in less extensive cytoreductive procedures because 37% (n=30) of 82 clinically suspect resected lesions did not contain malignant cells".

Sensitivity and specificity are not reported because this phase I study is not adequately powered to calculate these values. We therefore believe that it would not be scientifically accurate to report sensitivity and specificity. An expansion cohort study is currently ongoing.

One idea ?

i) Include normal cases – as patient operated for a small colon cancer lesion to be certain that normal tissues did not have auto fluorescence. The antibody could have a spatial distribution affected by cancer location.

That question is important because authors reported false positive at 20% of cases, on the result section.

This is an interesting suggestion for future work. We agree that some cases of false positivity could be attributed to auto-fluorescence of tissue. Also, the EPR effect could result in aspecific tracer accumulation, especially in the highest dose cohort. However, we see that the majority of tumor negative lesions do not show fluorescence (n = 30), and that there is a plausible explanation for almost all false positive lesions. Furthermore, we observed very low (auto)fluorescence in resected normal tissue surrounding the tumor, suggesting that autofluorescence of normal tissue is relatively low.

There were 10 false positive lesions detected in the 10 and 50 mg cohorts combined. Two of these lesions overexpressed CEA (1 due to colitis and 1 due to the presence of CEA-containing acellular mucin in a mucinous adenocarcinoma). In these two cases the uptake is specific due to targeting of the CEA. One lesion partially consisted of liver tissue, which was fluorescent as a result of hepatic clearance of our tracer. The remaining 7 lesions consisted of fibrotic inflamed lesions and reactions to foreign bodies. We believe that autofluorescence of these lesions occurs as a result of these reactions. We do not expect that this is aspecific tracer uptake as a result of the enhanced permeability and retention effect, because the lesions were not considered positive with gamma probe detection.

That paper is very interesting and had to be published

- 1 Marshall, M. V., Draney, D., Sevick-Muraca, E. M. & Olive, D. M. Single-dose intravenous toxicity study of IRDye 800CW in Sprague-Dawley rats. *Molecular imaging and biology : MIB : the official publication of the Academy of Molecular Imaging* **12**, 583-594, doi:10.1007/s11307-010-0317-x (2010).
- 2 Rijpkema, M. *et al.* SPECT- and fluorescence image-guided surgery using a dual-labeled carcinoembryonic antigen-targeting antibody. *Journal of nuclear medicine : official publication, Society of Nuclear Medicine* **55**, 1519-1524, doi:10.2967/jnumed.114.142141 (2014).
- 3 EMA. the EMA directives in: Radiopharmaceuticals based on Monoclonal Antibodies Directives 65/65/EEC, 75/318/EEC as amended, Directive 89/343/EEC, 3AQ21A).
- 4 Linssen, M. D. *et al.* Roadmap for the Development and Clinical Translation of Optical Tracers Cetuximab-800CW and Trastuzumab-800CW. *Journal of nuclear medicine : official publication, Society of Nuclear Medicine* **60**, 418-423, doi:10.2967/jnumed.118.216556 (2019).

REVIEWERS' COMMENTS

Reviewer #1 (Remarks to the Author):

The authors have adequately addressed the raised questions and discussion points.

I have no further remarks to the authors.

Reviewer #3 (Remarks to the Author):

the authors need to put the following INTO the actual paper, not just for the reviewer benefit.

What type of non-human toxicity testing has been done for the current conjugated antibody?

Together with the toxicity and safety tests performed for similar mAb conjugates (e.g. bevacizumab-IRDye800CW, cetuximab-IRDye800CW), and our own preclinical results with this specific tracer²

, the Dutch competent authorities waived the need for further toxicity testing^{3,4}

Point to point reply to the REVIEWERS' COMMENTS

Reviewer #1 (Remarks to the Author):

The authors have adequately addressed the raised questions and discussion points. I have no further remarks to the authors.

We would like to thank the author for the kind remarks and excellent feedback.

Reviewer #3 (Remarks to the Author):

the authors need to put the following into the actual paper, not just for the reviewer benefit. What type of non-human toxicity testing has been done for the current conjugated antibody?

Together with the toxicity and safety tests performed for similar mAb conjugates (e.g. bevacizumab-IRDye800CW, cetuximab-IRDye800CW), and our own preclinical results with this specific tracer², the Dutch competent authorities waived the need for further toxicity testing^{3,4}

These specific sentences on toxicity testing have been added to the methods section under the section Safety & pharmacokinetics (line 495 to 500). We would like to thank the author for the kind remarks and excellent feedback.